# Effect of Asymmetric Feathering Angle on the Aerodynamic Performance of a Flyable Bionic Flapping-Wing Rotor

**Si Chen [1], Le Wang [2,\*], Shijun Guo [3], Mingbo Tong [4], Yuanyuan He [5] and Jie Hu [6,\*]**

[1] College of Mechanical and Electrical Engineering, Wenzhou University, Wenzhou 325035, China
[2] School of Intelligent Manufacturing, Huzhou Vacational & Technical College, Huzhou 313099, China
[3] School of Aerospace, Transport and Manufacturing, Cranfield University, Cranfield MK43 0AL, UK
[4] College of Aerospace Engineering, Nanjing University of Aeronautics and Astronautics, Nanjing 210016, China
[5] School of Aerospace Engineering, Beijing Institute of Technology, Beijing 100081, China
[6] College of Engineering, Jiangxi Agricultural University, Nanchang 330045, China
\* Correspondence: 20200315@wzu.edu.cn (L.W.); hujie9@nuaa.euu.cn (J.H.)

**Abstract:** The current study involves an experimental as well as numerical study on the aerodynamic behavior of a flapping-wing rotor (FWR) with different feathering amplitudes ($-20°$–$50°$, $-50°$–$20°$, and $-35°$–$35°$). In order to fulfil the experimental test, an FWR which weighs 18.7 g is designed in this manuscript. According to the experimental and numerical results, it was observed that, compared with the cases under a zero average stroke angle, the cases under a positive average stroke angle or negative average stroke angle share a higher rotary speed given the same input voltage. Despite the fact that the negative average stroke angle would facilitate the generation of a higher rotary speed, the negative average stroke angle cases tend to generate the smallest lift-to-power ratio. On the other hand, the cases with a positive average stroke angle tend to share the largest lift-to-power ratio (about 1.25 times those of zero average stroke angle cases and about 1.6 times those of negative average stroke angle cases). The above study indicates that the application of a positive average stroke angle can provide an effective solution to further increase the aerodynamic performance of a bio-inspired FWR.

**Keywords:** bio-inspired robots; asymmetric feathering amplitude; flapping-wing rotor; lift efficiency

## 1. Introduction

Insects and birds have provided significant inspiration for the development of flapping-wing micro aerial vehicles (FWMAV) and this bio-inspiration gives the performance of stealth, agility, and maneuverability to the FWMAV, which is of paramount significance in the reconnaissance of hostile environments [1–4]. Thus, the aerodynamic and flight dynamic performance of the FWMAV has garnered significant interest and attention over the past several decades [5–7]. Among the studies for the flapping-wing system, symmetric feathering amplitudes are used in some of the research. For instance, Lua et al. [8] investigated the aerodynamics of tandem-configured 2D flapping wings in forward flight through both experimental and numerical methods. The results show that when the distance of the two wings is set as two times the chord length, the thrust of the tandem wings outperformed the combined thrust of two independently flapping single wings for phase angles ranging from approximately $-90°$ to $90°$. Wu et al. [9] presents a multidisciplinary experimental method to investigate a flapping wing's aero-elasticity and thrust production. The results show that, under a specific deformation phase and amplitude, the interaction between aerodynamic loads and both the bending and twisting motion can improve wing performance.

However, the symmetric feathering amplitudes of a real insect or bird should not be exactly symmetric. Thus, there is also much research focusing on the aerodynamic

behavior of the FWMAV under asymmetric feathering amplitudes [10,11]. For example, Rongfa et al. [12] presented an analysis of the kinematics of a flapping membrane wing based on the experiment. They found that the ornithopter has a higher upper-range stroke angle ($+30°$) while the flexing of the wing results in the lower-range stroke angle of $-20°$. Zhu and Zhou [13] carried out a numerical analysis on the aerodynamic behavior of a 2D flapping wing with an asymmetric stroke angle in both hovering and forward flight. They found that the aerodynamic performance of the wing at a low Reynolds number can be enhanced under an appropriate asymmetric stroke.

By combing the rotary motion of a rotorcraft and the flapping motion of a bird, a novel flapping-wing rotor (FWR) configuration was proposed [14,15]. Unlike the insect-like flapping-wing system, the flapping-wing rotor (FWR) consists of a passive rotary motion which is caused by the reverse Karman vortex. The speed of the rotary motion is influenced by the flapping motion, twisting motion, as well as the wings' deformation, and can greatly affect the FWR's lift generation. This kind of configuration is capable of short-range take-off or landing and exhibit higher aerodynamic efficiency than insect-like FW when the Reynolds number ranges from 2600 to 5000 [16,17]. In the past 10 years, a number of FWR models have been proposed. Guo et al. [15] proposed a piezoelectric actuator as well as a multi-bar of flexure hinges to drive an FWR. Dong et al. [18] designed an FWR with three wings and developed certain control mechanisms to control its pose. The hovering flight test shows their FWR can counteract the rotary-caused moment and maintain the attitude of their FWR.

As declared before in this paper, although some of the research has involved the asymmetric feathering amplitudes for a flapping-wing system (such as Robobee) [19], few research involving in the asymmetric feathering amplitudes for an FWR system is found. Compared with a flapping-wing system, an FWR configuration shares a unique passive rotary motion caused by the reverse Karman vortex. Moreover, a different average stroke angle leads to a different speed of the rotary motion and thus leads to a different lift. According to the past studies, the effect of asymmetric feathering amplitudes on its lift and rotary speed remains unknown.

In this work, an FWR has been proposed and fabricated to investigate the influence of asymmetric feathering amplitudes during the upstroke and downstroke on the FWR's aerodynamic forces. Additionally, a multi-body dynamics model as well as a CFD model of FWR are built up to calculate the FWR's numerical inertial forces as well as the FWR's numerical lift, assisting in the understanding of the experimental results. Then, the effect of the asymmetric feathering amplitudes on the FWR's lift and efficiency was studied by comparing the FWR's lift for the asymmetric feathering amplitudes cases with those for the symmetric feathering amplitudes cases.

## 2. Structure Design and Numerical Method

### 2.1. Structure of the FWR Model

As shown in Figure 1, a flapping-wing rotor (FWR) model with variable rigid twisting angle was designed and manufactured. Most 3D-printed components of the FWR model (in black) as shown in Figure 1a were made of high-performance nylon and the black rods were made of carbon/epoxy composite. At the same time, the wings' membrane is made from 12.5 μm polyimide film. The customized coreless motor shares 7.4 V rated voltage, 15 W rated power, and 30000 rpm rated speed respectively. Through the double-reduction gear unit, the motor provides force for pushing the pushrod while the reciprocating motion of the pushrod would generate the stroke motion. Due to the reverse Karman vortex, the stroke motion would also produce axial-symmetric lift and thrust on both wings as shown in Figure 1. Then, compared with a typical flapping-wing system such as Delfly [20], the two thrusts of the FWR in opposite direction form a moment that leads to a unique rotary motion ($\psi$). Although the FWR uses an ordinary crank-slider mechanism to realize its stroke motion, it is noted that the angle limiter and the stop blocks are specially designed to realize the variation of the rigid twisting angle ($\theta$) ranging from $10°$ to $50°$ during the

flapping process of the FWR without extra active control. Then, the total twisting angle can be divided into two parts, that is, the rigid twisting motion part caused by the aerodynamic forces and inertial forces acting on the wings' surface, as well as the twisting motion part due to the flexibility.

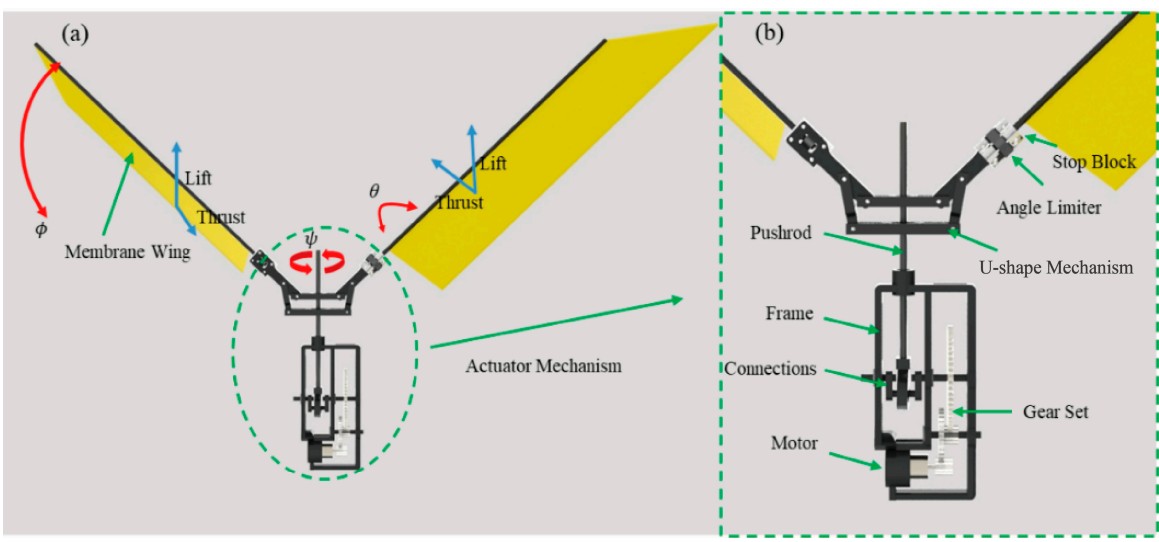

**Figure 1.** (**a**) Degree of freedom; (**b**) actuator mechanism of the FWR.

According to the Table 1, the total weight of the FWR is 18.7 g wherein the motor accounts for 5.5 g of the weight. Apart from the motor, the frame, the wing assembly, and U-shape mechanism weighs 2.8 g, 3.4 g, and 2.95 g, accounting for 15%, 18.2%, and 15.8% of the FWR's total weight. The weight of other parts including the bearings, pushrod, box, gear set, and connections are relatively small.

**Table 1.** The weight of the FWR's components.

| Component Name | Weight (g) | Quantity |
|---|---|---|
| Frame | 2.8 | 1 |
| Box | 0.5 | 1 |
| Motor | 5.5 | 1 |
| Gear set | 1.6 | 1 |
| Bearings | 0.25 | 5 |
| Pushrod | 0.2 | 1 |
| Connections | 0.25 | 2 |
| U-shape, echanism | 2.95 | 1 |
| Wing assembly | 1.7 | 2 |
| Total weight | 18.7 | 15 |

As depicted before, although the rotational speed of the coreless motor is as high as 30,000 rpm, the output torque is only about 4.78 mN·m. Thus, in order to provide enough thrust to push the pushrod, three gears are used to form the double-reduction gear unit as shown in Figure 2, wherein the addendum circle diameter of these three gears are $D_a$ = 29 mm, $D_b$ = 14 mm, and $D_c = D_d$ = 5 mm respectively. The depth of the teeth ($D_t$) is 1 mm. By assembling these gears properly, a double-reduction gear unit is designed with its gear ratio equal to 22.75:1. Moreover, the distance between the two holes on the connections is 5 mm which limits the feathering amplitude of the pushrod to ±5 mm.

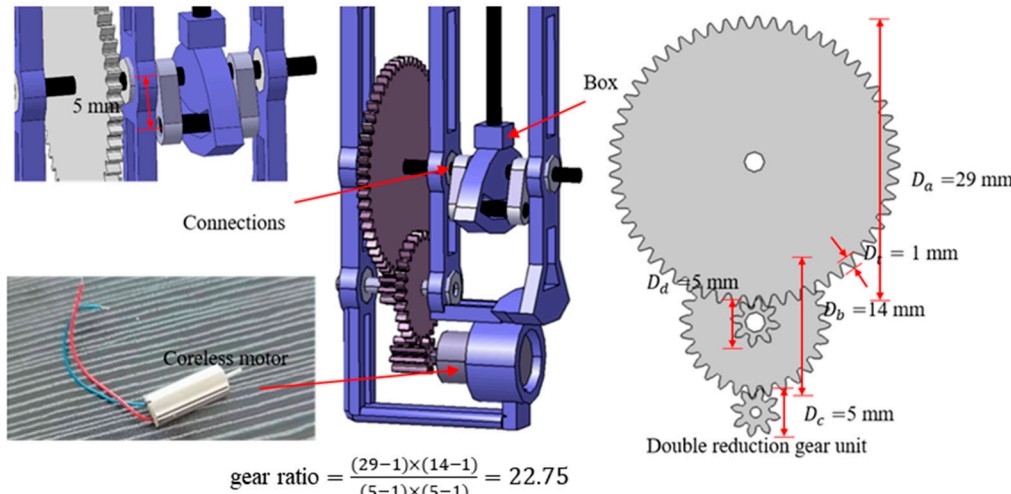

$$\text{gear ratio} = \frac{(29-1)\times(14-1)}{(5-1)\times(5-1)} = 22.75$$

**Figure 2.** Schematic diagram of the reduction gear set and actuator mechanism.

The wing assembly composed of stop blocks, angle limiter, leading-edge beam, subsidiary beams, and wing membrane is shown in Figure 3a, wherein both the stop blocks and angle limiter are made from nylon plastic. The leading-edge beam is made from 1.5 mm carbon/epoxy rod (0.5 g) while the other subsidiary beams are made from 1 mm carbon/epoxy rod (0.4 g). Moreover, the weight of the wing membrane is 0.4 g. The length of the spanwise beam and chordwise beam is 145 mm and 90 mm, respectively. In fact, with the wing membrane, the force obtained by the load cell should be the total force composed of inertial forces and aerodynamic forces [16]. In order to strip out the inertial component from the total forces, it is necessary to test the inertial component by testing the vertical forces generated by the FWR's model without the wing membrane [21,22]. To remove the inertial component from the total forces tested based on the original wing assembly model in Figure 3a, an additional inertial model is manufactured by removing the wing membrane as given in Figure 3b. The shape of the wing planform is not bio-inspired in this study.

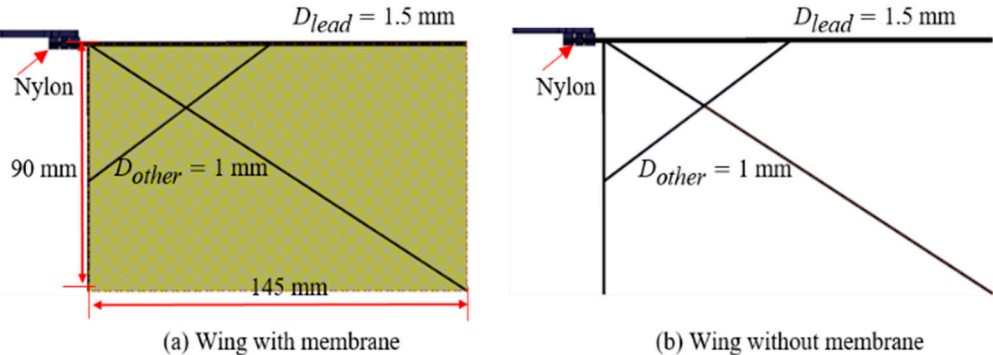

**Figure 3.** Geometric dimensions of the wing planform (with or without wing membrane) for the FWR.

### 2.2. The Setup of Experiment

In order to test the flapping kinematics of the flapping-wing rotor, it is significant to build up certain measurement system wherein the rotary speed of the flapping-wing rotor is assumed to be a constant value [23–25]. Since the variation of rotary speed within one single flapping period is neglected, it is practicable to measure the average rotary speed of the FWR in revolutions per minute (rpm) by videogrammetry. Moreover, by using the digital tachometer (DM6230), the flapping frequency of the FWR can be simply obtained. The detailed demonstration for the measurement system was given in our previous study [25].

Since the crank-slider mechanism is used to drive the flapping-wing rotor, the stroke angle curves nearly follow a sinusoidal function ($-20°$–$50°$, $-35°$–$35°$, or $-50°$–$20°$) as shown in Figure 4. The angle limiter and stop blocks can make the FWR generate passive twisting angle ($\theta$) varying from $10°$ (during the downstroke) to $50°$ (during the upstroke). The average twisting angle is chosen to be $30°$ since it is the optimum angle to achieve high lift for the rectangle wing [25]. It is noted that the phase difference between the stroke angle and twisting angle is nearly $90°$ as found in the previous studies [23–25]. The slope of the rotary angle ($\psi$) curve is assumed to be a constant value and obtained by experiment. It is noted that the Figure 4 just shows the rigid kinematics defined by the FWR structure and the time it takes to change the twisting angle from $50°$ into $10°$ is assumed to be 0.1 flapping period.

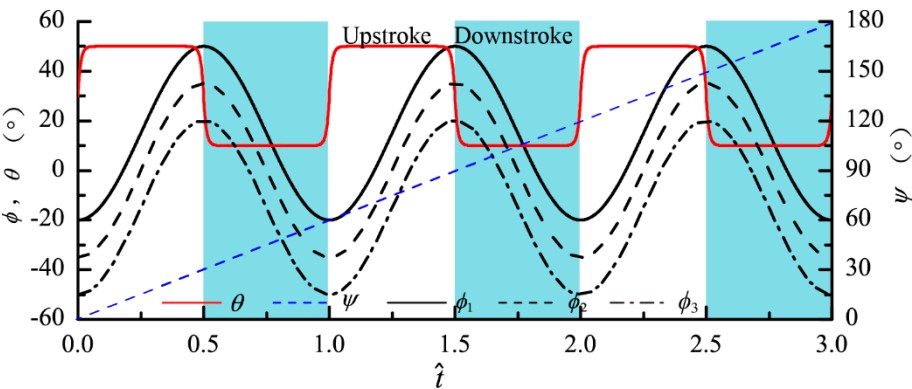

**Figure 4.** Kinematics of the FWR (not including the terms caused by the wings' deformation).

As shown in Figure 5, in order to realize the lift measurement, it is feasible to use a load cell (SEEED STUDIO 314990000) and a signal acquisition card (NI USB-6008). The 2nd-order low-pass Chebyshev algorithm in the LabVIEW code is employed to filter the measured voltage signals [26,27]. The electrical power consumption is obtained by multiplying the average current displayed on the power supply with the input voltage. The detailed introduction for the setup of the lift measurement system and the accuracy of the lift measurement system were discussed in our previous study [25].

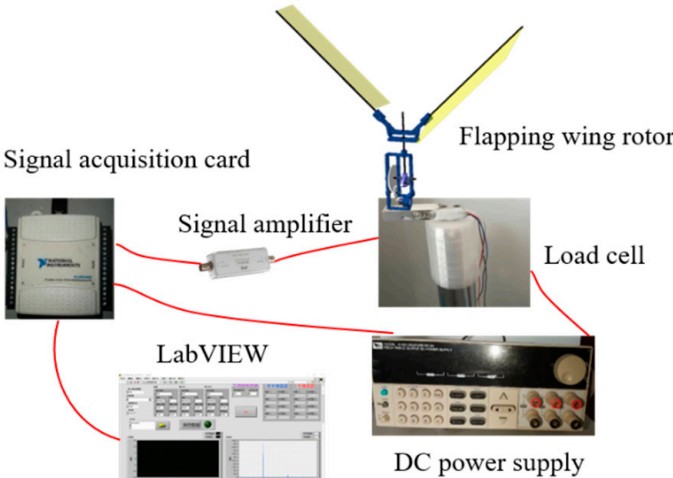

**Figure 5.** Lift-testing system for the FWR.

### 2.3. Numerical Model

In this manuscript, a comparison between the numerical and experimental results is performed. The CFD solver used is StarCCM+ 2020 and the mesh is also plotted by StarCCM+ software as given in Figure 6. The lift calculation in this study is fulfilled by

solving the unsteady Reynolds average equation built in StarCCM+ software. To ensure the numerical stability of the calculation, implicit Euler time discretisation method is applied. It is noted that the deformation of these wings is not included in this model. The height of the first mesh cell off the wall is set as 0.1 mm, and 10-layer boundary layer mesh is plotted. The total number of the meshes is about 2 million, and polyhedral meshes are used in the CFD calculation. The whole fluid field can be mainly partitioned into two regions: the background mesh region and the overset mesh region. A user-defined function is programmed in Python language to make the overset mesh region fulfill the rotation motions of the wings. Moreover, the $k - \omega$ SST turbulence model is applied as that used in previous study [28].

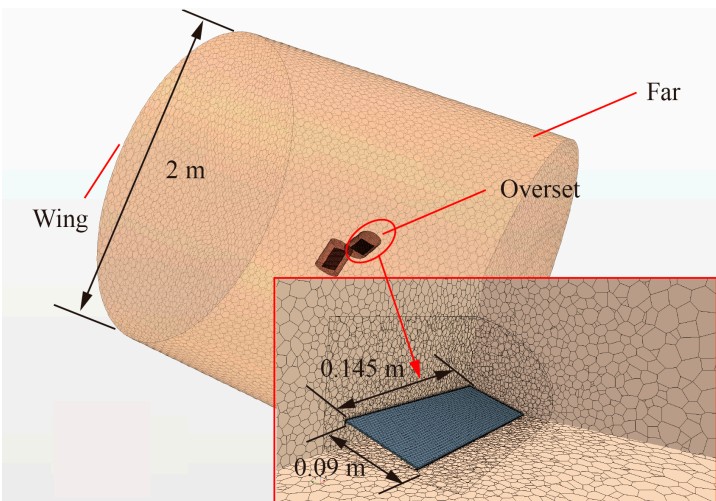

**Figure 6.** CFD model of the flapping-wing rotor.

For validating the CFD model in this paper, the lift and moment results by our CFD simulation are compared with those obtained by Wang et al. [24]. The variation function of twisting angle ($\theta$), stroke angle ($\phi$), and rotary angle ($\psi$) are set the same as those in the literature. According to Figure 7, it is observed that, by using our CFD model, the $\overline{C_L}$ equals to 0.88, and $\overline{C_m}$ equals to 0.145, which is quite close to those given by Wang et al. ($\overline{C_L}$ = 0.92 and $\overline{C_m}$ = 0.13). The difference should partly lie in the different mesh type and solver used. Generally, the above comparison illustrates that the CFD model in this paper is appropriate for the aerodynamic calculation of the FWR.

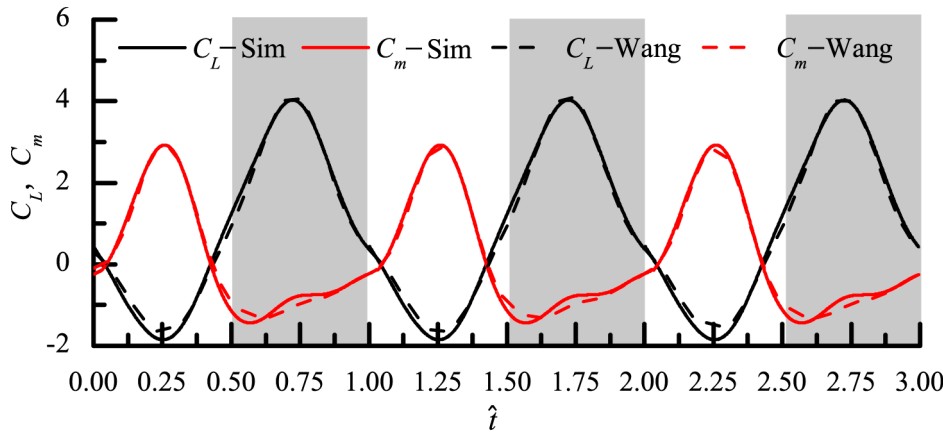

**Figure 7.** Comparison of lift coefficients and rotary moment coefficients between the CFD calculation by StarCCM+ software and the previous computational results.

To facilitate the understanding of the inertial force results tested by the experiment, an ADAMS model is also built up as shown in Figure 8. As depicted in Table 1 and Figure 3, the single wing assembly weighs 1.7 g and can be further decomposed into several parts, including the hot-melt adhesives (0.2 g), nylon stop blocks & angle limiter (0.2 g), main beam (0.5 g), wing membrane (0.4 g), and subsidiary beam (0.4 g). To keep consistency with the experiment, the wing membrane part should be removed from the ADAMS model upon calculating the inertial forces of the FWR encountered during its flapping process.

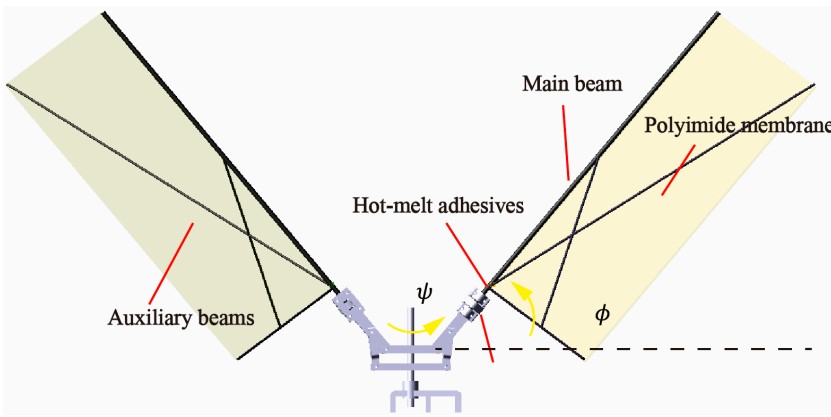

**Figure 8.** ADAMS model of the flapping-wing rotor.

### 3. Experiment Results and Discussion

*3.1. Measuring Cases*

As described in the previous subsection, a pair of wing assemblies are manufactured in this paper. The twisting angle of the flapping-wing rotor is set as from 10° to 50°. The average twisting angles are selected to be 30°. This is because, when the twisting angle maintains at 30° during the flapping process, the FWR share the highest lift efficiency [17,25]. Thus, in order to study the effect of the stroke angle on the lift generation for the flapping-wing rotor, the number of cases under the combination of different input voltage ($V_{input}$) and stroke angle ($\phi$) are listed in Table 2. The maximum voltage in these cases was 6.5 V since a higher voltage may lead to the stress concentration in the FWR's structure and wreck the structure. In order to ensure that the effect of the asymmetric stroke angle on the FWR's lift is easier to be observed, it is necessary to set the absolute value of the average stroke angle as large as possible in Cases 1 ($\phi \in [-20, 50]$) and Cases 3 ($\phi \in [-50, 20]$). However, the structural interference occurs at the U-shape mechanism of the FWR (as shown in Figure 1) when the stroke angle reaches about 53°. Thus, the stroke angles are set as −20°–50° and −50°–20° in Case 1 and Case 3.

**Table 2.** Test cases with different parameter and data setting.

| Cases 1 | $\phi$ | $V_{input}$ | Cases 2 | $\phi$ | $V_{input}$ | Cases 3 | $\phi$ | $V_{input}$ |
|---|---|---|---|---|---|---|---|---|
| 1-1 | −20°–50° | 3 | 2-1 | −35°–35° | 3 | 3-1 | −50°–20° | 3 |
| 1-2 | −20°–50° | 3.5 | 2-2 | −35°–35° | 3.5 | 3-2 | −50°–20° | 3.5 |
| 1-3 | −20°–50° | 4 | 2-3 | −35°–35° | 4 | 3-3 | −50°–20° | 4 |
| 1-4 | −20°–50° | 4.5 | 2-4 | −35°–35° | 4.5 | 3-4 | −50°–20° | 4.5 |
| 1-5 | −20°–50° | 5 | 2-5 | −35°–35° | 5 | 3-5 | −50°–20° | 5 |
| 1-6 | −20°–50° | 5.5 | 2-6 | −35°–35° | 5.5 | 3-6 | −50°–20° | 5.5 |
| 1-7 | −20°–50° | 6 | 2-7 | −35°–35° | 6 | 3-7 | −50°–20° | 6 |
| 1-8 | −20°–50° | 6.5 | 2-8 | −35°–35° | 6.5 | 3-8 | −50°–20° | 6.5 |

*3.2. The Effect of Asymmetric Stroke Angles on the Motion and Efficiency of the Flapping-Wing Rotor*

Based on the measuring method depicted in Section 2.2, the flapping frequency ($f_{flap}$), rotary speed ($\omega_r$), lift-to-power ratio ($L_{avg}/P_{input}$, average lift over input power of the

motor), and peak inertial forces for the FWR are tested as shown in Figure 9. According to Figure 9, despite the lift-to-power ratio going down with the increase of the input voltage, the flapping frequency, rotary speed, and peak of the inertial forces are positively correlated with the input voltage.

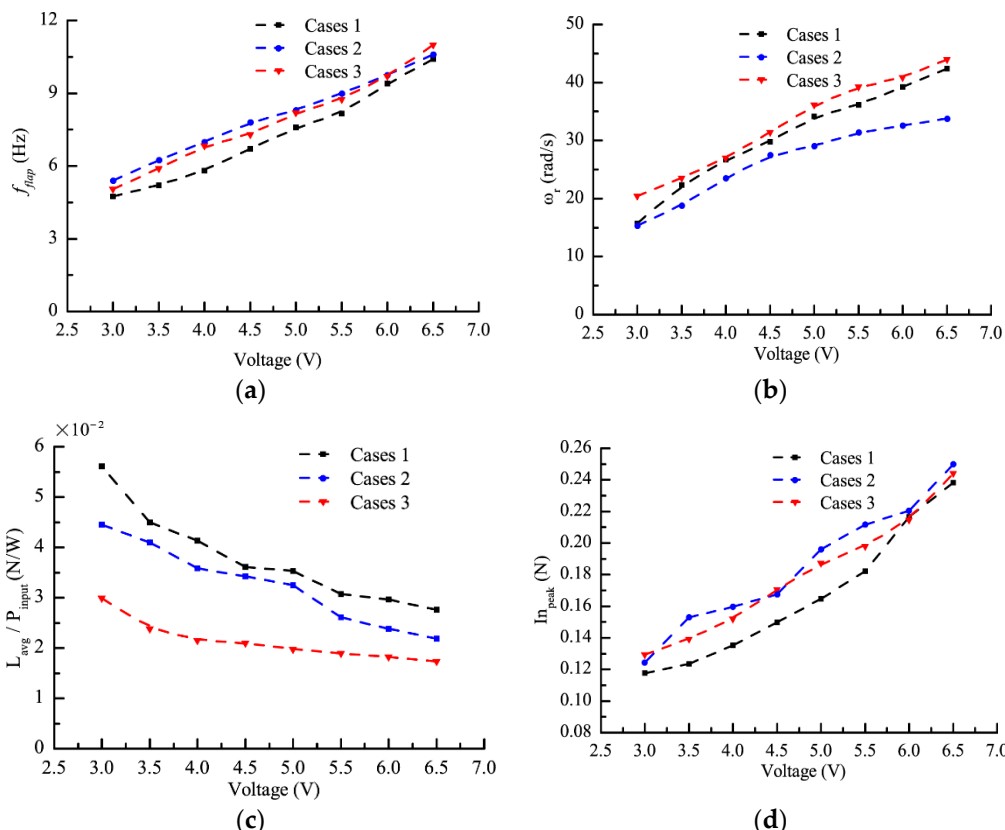

**Figure 9.** (**a**) Flapping frequency; (**b**) rotary speed; (**c**) lift-to-power ratio; and (**d**) peak values in the inertial forces' curves of the flapping-wing rotor under different voltage.

According to Figure 9a, similar flapping frequencies are found for Cases 1 ($\phi \in [-20, 50]$), Cases 2 ($\phi \in [-35, 35]$), and Cases 3 ($\phi \in [-50, 20]$) under a different input voltage in general. Thus, the peaks of the inertial forces under these cases also stay at the same level as shown in Figure 9d. Compared with the cases in Cases 2 ($\phi \in [-35, 35]$), the cases in Cases 1 ($\phi \in [-20, 50]$) or Cases 3 ($\phi \in [-50, 20]$) share a relatively higher rotary speed given the same input voltage. The rotary speed for Cases 3-8 (7 rev/s) especially are 1.3 times larger than that for Case 2-8 (5.375 rev/s). Among the cases in Cases 1 ($\phi \in [-20, 50]$), Cases 2 ($\phi \in [-35, 35]$), and Cases 3 ($\phi \in [-50, 20]$), it is found that the positive average stroke angle in Cases 1 ($\phi \in [-20, 50]$) leads to the largest lift-to-power ratio (about 1.25 times those in Cases 2 and about 1.6 times those in Cases 3). The negative average stroke angle tends to generate the smallest lift-to-power ratio despite the negative average stroke angle facilitating the generation of a higher rotary speed.

### 3.3. Experimental and CFD Simulation Comparison

From the measuring results in Figure 9a, it can be found that Case 1-6, Cases 2–5, and Cases 3–5 share a similar flapping frequency (around 8.2 Hz). Then, the three cases are emphasized as the typical cases, and the total force (T), the inertial force (I), and the aerodynamic lift forces (L) in the three cases are shown in Figure 10. The force–time curves of the three cases start at the beginning of the upstroke when the maximum negative inertial force is found. An about 90° phase difference between the inertial force (I) and lift force (L)

is observed since the minimum vertical acceleration occurs at the mid-downstroke while the aerodynamic lift nearly reaches the maximum value at this moment.

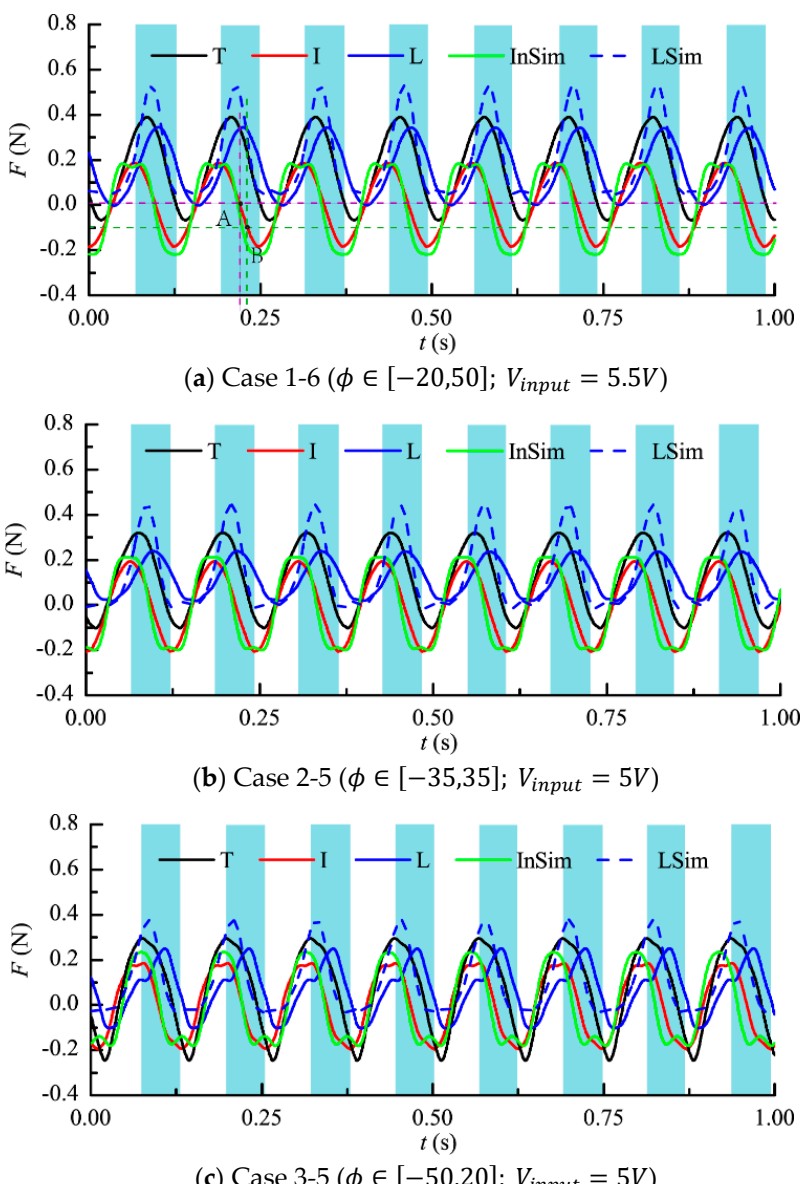

(**a**) Case 1-6 ($\phi \in [-20,50]$; $V_{input} = 5.5V$)

(**b**) Case 2-5 ($\phi \in [-35,35]$; $V_{input} = 5V$)

(**c**) Case 3-5 ($\phi \in [-50,20]$; $V_{input} = 5V$)

**Figure 10.** Total force, inertial force, and lift force of the FWR.

From Figure 10a, for Case 1-6, the peak values of the total force (T), inertial force (I), and lift force (L) are around 39 g, 18.6 g, and 34.5 g, respectively, wherein the average inertial force throughout the whole flapping period is very close to zero as it should be, theoretically. Although significant negative total force occurs during the upstroke, the negative lift is nearly zero. This means the instantaneous negative total force results from the negative inertial force and the rotary motion has already eliminated the negative lift during the whole flapping period. At the same time, it is noted that the stroke angle goes to 15° at the mid-downstroke (as marked by point A) instead of 0° (as marked by point B) in Figure 10a. At the mid-downstroke, the wings' flapping velocity reaches its peak which leads to the largest normal forces perpendicular to the wings' surface. However, the lift is the vertical component of the normal forces at this moment. It is not until the time marked as point B that the lift equals to the normal forces and the flapping velocity is smaller than the peak it can reach. Thus, the maximum lift occurs between point A and point B when

the flapping velocity is close to the peak it can reach and the stroke angle is close to $0°$ so that the lift is close to the normal forces.

As for Case 2-5, the peak values of the total force (T), inertial force (I), and lift force (L) are around 0.32 N, 0.193 N, and 0.239 N, respectively. A similar variation trend is found for the force curves in Case 1-6 and Case 2-5. Although a different average stroke angle is found in Case 1-6 and Case 2-5, they share a similar peak inertial force due to the similar flapping frequency. Meanwhile, the ratio between the peak of the lift force and inertial force is 1.2 in Case 2-5, which is smaller than that in Case 1-6 (1.73). The average lift in Case 1-6 (0.16 N) is found to be higher than that in Case 2-5 (0.119 N) by 35%.

When it comes to Case 3-5, it is found that there exists an apparent negative instantaneous lift force during the upstroke (about $-0.1$ N), accounting for about 42% of the peak positive lift force (about 0.24 N) during the downstroke. Thus, the average force of the FWR decreases to an even lower value (0.07 N). Moreover, in both Figure 10a–c, similar peaks of the inertial forces are found in Case 1-6, Case 2-5, and Case 3-5.

According to Figure 10, the variation trend of the inertial forces' curves from the ADAMS simulation is close to those from the experiment despite there being some small bumps during the end of the upstroke and the end of the downstroke. Since the stroke angle nearly follows a sinusoidal function, at the end of the upstroke and the end of the downstroke, the magnitude of the stroke acceleration reaches the peak and the changing rate of the stroke acceleration is relatively small. However, the larger stroke angle leads to a smaller vertical acceleration component and thus leads to the small bumps for the InSim curves in Figure 10. The phenomenon is more apparent during the end of the downstroke in Case 3-5 and during the end of the upstroke in Case 1-6. From Figure 10a, the peak value of the simulated inertial forces and the experimental inertial forces is $-0.185$ N and $-0.218$ N, respectively. The peak differences in Figure 10b ($-0.206$ N and $-0.187$ N) and Figure 10c ($-0.196$ N and $-0.181$ N) become smaller.

When it comes to the lift results, it is observed that the variation trend of the lift forces' curves from the CFD simulation is close to that from the experiment. This difference between our CFD results and experiment results should be mainly due to the fact that the wings' deformation is neglected in our CFD model.

*3.4. The Effect of Asymmetric Stroke Angles on the Average Lift Generation*

Despite the fact that the motion and efficiency parameters of the FWR under different input conditions have been given in Section 3.3, it is also important to measure the average lift for these cases. The average lift for Case 1 ($\phi \in [-20, 50]$), Cases 2 ($\phi \in [-35, 35]$), and Cases 3 ($\phi \in [-50, 20]$) are then all shown in Figure 11. With the increase of the input voltage, both the total force and lift increase rapidly despite the average inertial force remaining nearly zero. Whatever the input voltage is, the average lifts in Cases 1 ($\phi \in [-20, 50]$), Cases 2 ($\phi \in [-35, 35]$), and Cases 3 ($\phi \in [-50, 20]$)) are about 1.25 times those in Cases 2 ($\phi \in [-35, 35]$) and about 1.5 times those in Cases 3 ($\phi \in [-35, 35]$). Apart from the cases in Cases 3 ($\phi \in [-50, 20]$), the average lifts show a nearly linear relationship with the input voltage. In Cases 3 ($\phi \in [-50, 20]$), the linear relationship only stands when the input voltage is over 4.5 V. From the weight parameters listed in Table 1, it is found that the FWR weighs 0.187 N while the maximum average lift (0.201 N) is achieved in Case 1-8 according to Figure 11. In this case, the lift-to-weight ratio reaches 1.07 and the take-off test is conducted in our previous study [29]. During the take-off test, a 4 g wire was used to connect an external power supply to the FWR model (not taking into account the battery weight). Although the positive average stroke angle can facilitate the lift generation according to the experiment results, the corresponding reason of this phenomenon remains unknown. In our next studies, a certain explanation for the phenomenon is planned to be given by focusing on the real variation curves of the rotary angle (rather than assuming that the rotary speed is a constant value as found in previous studies [23–25]).

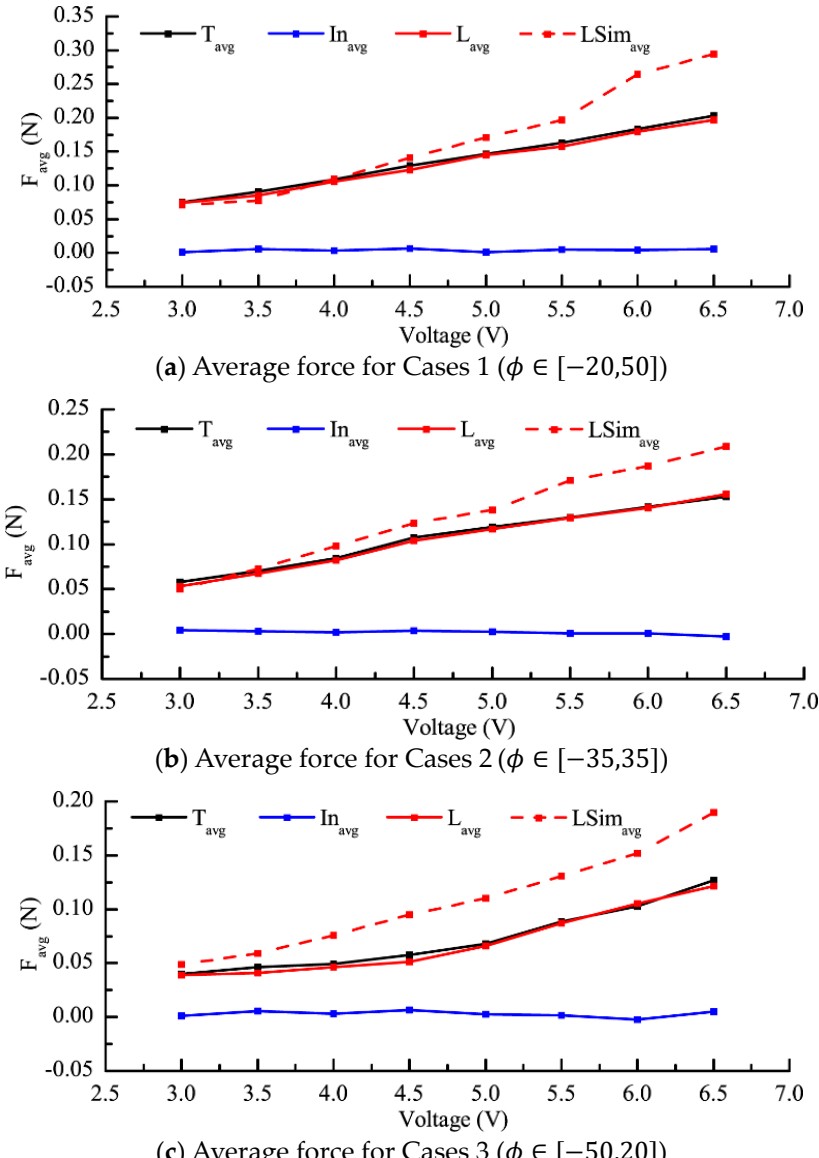

**Figure 11.** Measurement of average force for the FWR.

In order to further confirm that the positive effect of the positive average stroke angle on the FWR's lift observed in the experiment is an inevitable result from the perspective of aerodynamics, the average lifts of the FWR are also calculated based on the CFD method by setting the same motion parameters as those measured by experiment. According to Figure 11, the average lifts from the simulation is larger than those from the experiment and the difference between the simulated average lift and experimental average lift becomes larger with the increase of the input voltage (the flapping frequency is positively correlated with the input voltage as shown in Figure 9). The maximum difference occurs in Case 1-8. In this case, the simulated average lift is larger than the experimental average lift by 49.4%. One of the reasons is that the deformation of the wings is relatively small when a small input voltage (<4 V) is applied while a greater deformation of the wings would be found with the increase of the input voltage. Another reason can be that the rotary speed is not a standard constant, although we make the same assumption as used in previous studies [23–25].

## 4. Conclusions

In this paper, a flapping-wing rotor model is manufactured with two-stage reduction gear. The gear ratio is set as 22.75:1. By using the angle limiter and the stop blocks, the flapping-wing rotor can realize a variable rigid twisting angle during the downstroke and upstroke. Then, an investigation is conducted into the effect of the average stroke angle on the lift, motion, and efficiency of the FWR by an experiment. In the experiment, by taking the zero average stroke angle cases as the baseline, the effect of the positive (negative) average stroke angle on the lift generation is studied. A CFD model is also built up to calculate the FWR's numerical lift to further demonstrate that the positive effect of the positive average stroke angle on the FWR's lift observed in the experiment is an inevitable result from the perspective of aerodynamics. From the experimental study, the following remarks are drawn:

- Compared with the cases in Cases 2 ($\phi \in [-35, 35]$), the cases in Cases 1 ($\phi \in [-20, 50]$) or Cases 3($\phi \in [-50, 20]$) share a relatively higher rotary speed given the same input voltage. The rotary speed for Case 3-8 (7 rev/s) are 1.3 times larger than that for Case 2-8 (5.375 rev/s).
- The positive average stroke angle in Cases 1 leads to the largest lift-to-power ratio (about 1.25 times those in Cases 2 and about 1.6 times those in Cases 3). The negative average stroke angle cases tend to generate the smallest lift-to-power ratio despite the negative average stroke angle facilitating the generation of a higher rotary speed.
- Through the time history curves of the lift in Case 1-6, Case 2-5, and Case 3-5, it is found that under the same flapping frequency, the peak inertial forces under different average stroke angles are similar with each other. Moreover, despite there being an apparent negative lift during the upstroke in Case 3-5 (negative average stroke angle), the instantaneous lift in Case 1-6 and Case 2-5 is positive or nearly zero.
- Whatever the input voltage is, the average lifts in Cases 1 ($\phi \in [-20, 50]$) are about 1.25 times those in Cases 2 ($\phi \in [-35, 35]$) and about 1.5 times those in Cases 3 ($\phi \in [-50, 20]$).
- Among all the cases in this paper, the highest lift-to-weight ratio reaches 1.07, which means the flapping-wing rotor can realize vertical take-off.

Above all, the positive average stroke angle will not only facilitate the increase of the rotary speed but also benefit the FWR's lift efficiency.

**Author Contributions:** Formal analysis, J.H. and Y.H.; Investigation, S.G.; Data curation, M.T.; Writing—original draft, S.C.; Writing—review & editing, L.W. All authors have read and agreed to the published version of the manuscript.

**Funding:** This research was funded by the Natural Science Foundation of Zhejiang Province (Grant No. LQ22A020006); the Science and Technology Project of Jiangxi Education Department (Grant No. GJJ200452); the Basic Public Welfare Research Program of Wenzhou (Grant No. G20210007); and The National Natural Science Foundation of China (Grant No. 11972079).

**Institutional Review Board Statement:** Not applicable.

**Informed Consent Statement:** Not applicable.

**Data Availability Statement:** Not applicable.

**Conflicts of Interest:** The authors declare no conflict of interest.

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
