# Peer review of "Effect of Asymmetric Feathering Angle on the Aerodynamic Performance of a Flyable Bionic Flapping-Wing Rotor"

_applsci, doi:10.3390/app13063884_

Round 1
Reviewer 1 Report
This paper presents a study on the asymmetric feathering angle on the aerodynamic performance of a flapping wing rotor. Numerous grammar problems, rusty English expression created significant barrier to understand the content of the paper. The paper is not written professionally; the whole paper is very hard to follow. Many symbols were not defined at all. Some examples are given below. There are some major technical problems in the paper body as well.
1. The purpose of the CFD simulation is not clear. The value provided by CFD for the understanding of the experimental results is very little.
2. The CFD simulation setup is obscure. No tests on the mesh quality is conducted. There is a difference of over 20% in the time-averaged moment coefficient comparison with the literature, which is not acceptable.
3. Why did the author pick the current three feathering amplitude setup? The reason for three setups is not clear.
4. The discussion of the results is lack of deep explanation for the phenomenon observed.
5. Was there any free-flight test conducted? The last conclusion is the flyer flyable. The author can simply let it fly to prove this. Was the free flight taking battery weight into account?
6. Some example of grammar problems, Line 17, It should be “through experimental and numerical….” Line 37, it should be “investigated the aerodynamics of …”, remove “on”; Line 45, It should be “…real insect or bird”. Just pick these three. There are much more than these.
Reviewer 2 Report
Review of Manuscript applsci-2265490: Effect of asymmetric feathering angle on the aerodynamic performance of a flyable bionic flapping wing rotor.
The paper pertains the numerical and experimental investigation of a flapping wing rotor. The topic is of interest in the ever evolving field of micro-aerial vehicles, the experiments and the numerical simulations appear to have been conducted properly, the paper is well presented and the results are interesting. However a few points need to be ironed out before publication.
Comments as follows:
1. line 16 - The authors refer to the flapping amplitudes as (-20 -50, -50 -20, -35 -35), but looking at figure 4 shouldn't they be (-20 50, -50 20, -35 35)? Please correct throughout the paper.
2. line 66 - "to control the its pose" - remove "the".
3. line 70 - Replace "little researchers" with "few researchers".
4. line 94 - Replace " would also produces" with "would also produce".
5. line 99 - I understand that the amplitude of the flapping and rigid twisting and how they are phased with respect to one another is crucial in order to obtain Lift. How were these parameter chosen? Please add considerations about this in the revised paper.
6. line 111-113 - The authors mention a reduction ratio of 22.75. I was at first baffled because by using the gears diameters I obtained 29*14/(5*5)=16.24, but then counting the number of teeth from figure 2 I obtained 22.75 too. Perhaps the authors should clarify this in the paper.
7. line 116 - Please replace "plotted" with "shown".
8. line 156 - Replace "filterer" with "filter"
9. lines 166-174. More details should be given about the numerical simulations. what equations were solved ( non-compressible URANS I suppose)?. What time discretisation was used ? etc. Please expand.
10. figure 9 and following - Please have all the sub-pictures of the same figure on the same page.
11. figure 9 and following - I wonder if, instead of referring to case 1, 2 and 3 in the picture captions and labels, the authors could refer directly to the flapping amplitude like "psi [-20 50]" etc.
12. figure 10. See point above, please expand captions, e.g. "Case 3-5; phi [50-20], V_input = 5V"
Reviewer 3 Report
The work aims to investigate the effect of asymmetric feathering amplitudes during the upstroke and downstroke on the flapping wing rotor aerodynamic forces. Moreover, a complex multi-body and CFD model of FWR are developed to calculate the numerical inertial forces and lift. Simulation results are discussed related to experimental tests.
Some new findings are highlighted about constructive versions such that the wing performance would be enhanced.
A particular research strategy has been approached, based on asymmetric feathering angle, which was validated through a multi-body dynamics model followed by a CFD model of FWR.
Following the study, the authors found a new constructive version of FWR showing that the application of the flapping kinematics with the positive average stroke angle can provide a feasible solution for aerodynamic improvement of a bio-inspired flapping wing rotor.
The improvements brought to the FWR system, the novelty and the meaningful results of the research are summarized in the conclusion paragraph which is consistent with the applied methodology.
I appreciate that the study starts from an updated state-of-the-art, based on current bibliographic references. I would recommend a better quality of the Figures 6 and 8, and a carefully checking of the text.
Finally, the manuscript can be considered for publication after minor revision.
I would recommend giving a clear picture of the lift testing system for the flapping wing rotor (Figure 5). Conclusions are in agreement with the experiment and simulation, and give a clear insight of the results obtained.
The manuscript is well written and, in my opinion, could be considered for publication after minor corrections.
Round 2
Reviewer 1 Report
My comments and concerns were addressed appropriately. The quality of the paper does not reach the standard for publication in this journal.